# Pharmacists’ Professional Satisfaction and Challenges: A Netnographic Analysis of Reddit and Facebook Discussions

**DOI:** 10.3390/pharmacy12050155

**Published:** 2024-10-12

**Authors:** Marius Călin Cherecheș, Hajnal Finta, Răzvan Mihai Prisada, Aura Rusu

**Affiliations:** 1Faculty of Pharmacy, “George Emil Palade” University of Medicine, Pharmacy, Science and Technology, 540142 Târgu Mures, Romania; hajnal.finta@umfst.ro (H.F.); aura.rusu@umfst.ro (A.R.); 2Faculty of Pharmacy, “Carol Davila” University of Medicine and Pharmacy, 020956 Bucharest, Romania; razvan.prisada@umfcd.ro

**Keywords:** pharmacists, pharmacy, professional satisfaction, netnography, professional recognition, workplace dynamics, healthcare professionals, qualitative research, pharmacists’ satisfaction

## Abstract

Pharmacists, essential healthcare providers, face significant challenges in professional satisfaction and well-being. This study investigates the factors influencing pharmacists’ professional satisfaction, mainly focusing on workload, organizational support, job autonomy, work–life balance, and resilience against burnout. Data were collected from relevant online forums on Facebook and Reddit using a netnographic methodology. The data were anonymized and thematically coded to identify key themes from 23 conversation threads, primarily involving or concerning Romanian pharmacists. The analysis revealed several critical issues: widespread dissatisfaction with salaries, challenges in professional recognition, and the demanding nature of university education. Additional themes included economic and financial insights, global trends and technological impacts, personal experiences and satisfaction, professional growth and education, regulatory and market environment, and workplace dynamics. Findings indicate these factors significantly impact pharmacists’ job satisfaction and overall well-being. The study concludes that addressing these issues through targeted interventions, such as policy reforms, educational updates, and enhanced organizational support, can improve the professional lives of pharmacists, thereby enhancing their contributions to healthcare outcomes.

## 1. Introduction

Pharmacists are crucial healthcare professionals in the healthcare system who play a vital role in providing patients with various essential services. These services include offering expert medication advice, ensuring proper medication use, educating patients about disease prevention, promoting health and wellness, and providing interventions to enhance patient outcomes. Despite their critical contribution, pharmacists often face significant challenges that impact their professional satisfaction and well-being [1].

Several key factors influence the professional satisfaction of pharmacists. Workload, including the volume and complexity of tasks, plays a significant role [2]. Organizational support, such as access to resources and effective teamwork, also impacts satisfaction. Additionally, job autonomy, or the freedom to decide and set priorities, can affect how satisfied pharmacists feel. Furthermore, balancing work and personal life is crucial for overall professional satisfaction. These factors contribute to pharmacists’ overall well-being and job satisfaction in their professional practice [3,4,5]. One study examining the organizational commitment of pharmacists in Lebanon revealed that factors such as older age, involvement in student training, and overall dissatisfaction with their careers significantly contributed to burnout [6]. Moreover, studies suggest that experiencing professional satisfaction and developing resilience are vital for mitigating the adverse consequences of job demands and minimizing the risk of burnout [7,8]. Furthermore, studies in the United States show that pharmacist resilience significantly predicts lower burnout and higher job performance [9].

The work environment significantly influences pharmacists’ organizational commitment and professional satisfaction. Studies highlighted that inadequate facilities, poor management practices, and lack of recognition increase stress and burnout among pharmacists [10,11,12,13]. For example, a study in Nigeria found that handling multiple tasks simultaneously, combined with insufficient support, resulted in high fatigue levels and decreased motivation among medical professionals, including pharmacists [14].

Burnout, a significant concern in the pharmacy field, manifests as emotional exhaustion, depersonalization, and a decreased sense of personal accomplishment. Research indicates that the high prevalence of burnout among pharmacists can be attributed to excessive workloads, insufficient staffing levels, inadequate support from their organizations, and the demanding nature of their roles. Burnout is a critical issue that needs to be addressed to ensure the well-being of pharmacists and the quality of patient care [10,15,16]. For instance, a survey conducted among community pharmacists in Vietnam revealed that those experiencing burnout had significantly lower scores in medication counselling activities compared to their non-burnout counterparts, indicating a direct negative impact on their professional performance [15]. Positive stress, or eustress, can stimulate work performance, but negative stress, or distress, can interfere with a worker’s physical, psychological, or social stability [11].

Understanding the factors contributing to professional dissatisfaction and burnout among pharmacists is essential for developing effective interventions. Increasing staffing levels, improving management practices, and providing better organizational support can help alleviate burnout and enhance job satisfaction [17,18,19]. Furthermore, fostering a supportive work environment and promoting work–life balance is critical for maintaining pharmacists’ well-being and professional commitment [20,21,22,23].

While the demand for pharmacists is steadily increasing due to the ageing population, expanding healthcare services, and the need for specialized medications, future students’ interest in pursuing a pharmacy degree has been a topic of concern in recent years. Research indicates that pursuing pharmacy as a study and career path involves personal interests, career prospects, and educational exposure. Woldekidan et al. (2020) found that having pharmacy as a first choice, leadership experience, and previous related work experience significantly impact students’ decisions to pursue postgraduate education in pharmacy [24]. Pharmacy education faced various challenges, such as the perceived difficulty of the program, concerns about job prospects, and competition with other healthcare professions. In some regions, such as Kuwait, the primary motivation for pursuing a pharmacy degree is the inability to gain admission to medicine or dentistry programs, which suggests that pharmacy may not be the first choice for many aspiring healthcare professionals [25].

Programs such as summer camps are vital in attracting students to the pharmacy field by offering hands-on learning and insight into the pharmacy curriculum and university community, as described by Rosenberg et al. (2023) [26]. Additionally, another study emphasizes that personal factors and career advice significantly influence students’ decisions to study pharmacy in Vietnam [27].

Netnography, a qualitative research method adapted for online environments, offers a unique approach to understanding pharmacists’ experiences and perceptions. By analyzing digital traces and user interactions in online forums and social media, researchers can gain in-depth insights into the social and cultural contexts that shape pharmacists’ professional satisfaction [28,29,30]. Netnography, a qualitative research methodology that utilizes online data from social media, forums, and other online platforms, is widely applied in diverse areas such as tourism and business. It offers a unique opportunity to delve into and understand consumer behavior, community dynamics, and interactions within online communities. Netnography showcased its significant potential in informing and shaping healthcare research, allowing researchers to gain valuable insights into patient experiences, health-related behaviors, and the impact of online communities on healthcare decisions [31,32,33].

This study aims to comprehensively understand Romanian pharmacists’ perceptions, experiences, and dissatisfaction regarding their professional satisfaction. Utilizing a netnographic methodology, it seeks to identify the key factors that influence the image of the profession and the degree of professional satisfaction. The insights gained from this research will be valuable in informing policies and practices to improve the work environment and overall satisfaction of pharmacists, ultimately contributing to better healthcare outcomes.

## 2. Materials and Methods

The first step entailed gathering data from relevant online forums and communities on the social media platform Facebook and forum social network Reddit. The researchers selected conversations from these platforms based on their suitability for the research objectives and the prevalence of discussions about the pharmacy profession. We analyzed 8 conversation threads from Facebook groups (from 2024) and 15 threads from Reddit (from 2019 to 2024).

Following data collection, the researchers anonymized the data to ensure the privacy and confidentiality of the participants. The anonymized data were then processed using the Delve Qualitative Analysis Tool, a qualitative analysis software, to facilitate the thematic analysis coding process [34]. We analyzed these 23 conversation threads and meticulously coded the comments using predefined codes and emergent codes identified during the analysis of the transcripts. This dual approach allowed us to capture both expected and unanticipated themes, providing a comprehensive understanding of the data. The coding reliability was ensured through double coding using the Delve Qualitative Analysis Tool.

The coded data were systematically categorized and contextualized, allowing the researchers to deeply understand the social, cultural, and behavioral dynamics within online communities. Through this process, we identified and grouped the data into 8 distinct categories, each highlighting different aspects of the pharmacy profession. This thorough analysis culminated in developing a substantive theory, providing a robust framework for interpreting and conceptualizing the phenomena observed in online social media space [35,36,37].

The study received Institutional Review Board (IRB) approval (no. 3088/22.04.2024), ensuring the ethical and responsible conduct of the research. All participant data were anonymized to protect privacy, and the study was conducted following ethical guidelines approved by the IRB decision, ensuring participant confidentiality.

## 3. Results and Discussion

The pharmacy field witnessed dynamic changes influenced by various factors, from economic pressures to technological advancements. To better understand the factors influencing professional satisfaction among pharmacists, we conducted a qualitative research study analyzing conversations from social media platforms. This study aims to identify and categorize the key themes emerging from pharmacists’ discussions, providing insights into their professional experiences and challenges.

We analyzed 23 conversation threads and meticulously coded the comments using predefined codes and emergent codes identified during the analysis of the transcripts. This dual approach allowed us to capture both expected and unanticipated themes, providing a comprehensive understanding of the data. After analyzing all the codes, we grouped them into eight categories during the contextualization phase of our research. Each category encompasses a set of attributes and descriptions that reflect different aspects of the pharmacy profession.

The categories identified are as follows, ordered alphabetically:Cultural Insights and Humor: this category captures observations about societal attitudes toward the pharmacy profession and informal expressions, such as humor and sarcasm, used by pharmacists.Economic and Financial Insights: this category discusses the economic factors affecting pharmacists, including salary expectations, the impact of the cost of living, and broader financial outcomes related to the profession.Global Trends and Technological Impacts: this category captures global trends affecting pharmacy practice and the impact of technological advancements such as AI on the profession.Personal Experiences and Satisfaction: this category reflects the personal experiences of pharmacists, encompassing career satisfaction, personal anecdotes, and emotional reactions to their professional roles.Professional Growth and Education: this category covers all aspects of educational background, continuous professional development, growth opportunities, and interdisciplinary collaboration within the pharmacy field.Professional Identity and Public Perception: this category examines how pharmacists are perceived both within the healthcare community and by society at large, including misconceptions, stereotypes, and the level of professional recognition.Regulatory and Market Environment: this category focuses on the regulatory, legal, and market-driven aspects impacting the practice and business of pharmacy.Workplace Dynamics: this category explores the dynamics of different working environments, challenges faced in the workplace, and the realities of the job market for pharmacists.

The categorization process resulted in the assignment of multiple codes to each category, as outlined in Table 1 below:

This structured categorization offers a comprehensive overview of the myriad factors impacting pharmacists’ professional satisfaction. Each category and its associated codes provide valuable insights into the diverse experiences and challenges faced by pharmacists, paving the way for targeted interventions and policy recommendations to enhance professional satisfaction and improve the overall practice of pharmacy.

Please note that the Romanian currency is Leu (plural: Lei) and the symbol is RON. Within the document, both forms are used interchangeably.

### 3.1. Cultural Insights and Humor

Social media discussions among pharmacists provide valuable insights into the challenges and coping strategies within the profession. This category explores the intersection of humor and frustrations, offering a nuanced understanding of pharmacists’ experiences. Analyzing the snippets within this category revealed nine key findings that emphasize public perceptions of pharmacists and humorous approaches to various situations (Table 2).

#### 3.1.1. Challenges in Choosing the Pharmacy Profession

Pharmacists often use humor to reflect on the challenges of their career choice with a mix of irony and resignation. For instance, one comment illustrates this sentiment:


*“Why would one pursue a career in pharmacy (domain) when you are no longer regarded as a healthcare professional but merely a salesperson? In 2020, starting salaries in large cities were as low as RON 2050 net (EUR 410), and depending on luck, you might or might not receive bonuses.”*


This quote highlights the perception that the pharmacy profession is devalued, comparing pharmacists to mere salespeople. It also underscores their financial challenges, especially at the beginning of their careers. Since the 2000s, the profession has shifted from preparing and dispensing medicine to a more patient-centered approach. As a result, the training requirements and public perception changed [38].

#### 3.1.2. Correct Spelling and Precision Are Essential Aspects of the Pharmacy Profession

The emphasis on accurate spelling and precision also brings out humor. The meticulous nature of the pharmacy profession, where even minor mistakes can have significant consequences, is often humorously highlighted. For example, a comment notes:


*“Step one is probably spelling Pharmacist correctly… and no offence, but based on your post, you probably have a ways to go before getting into a pharmacy program.”*


The humorous remark emphasizes the importance of precision in pharmacy while gently reprimanding those thinking about entering the profession.

#### 3.1.3. Comparison with the Professor from “Breaking Bad”

The qualifications and skills of pharmacists are humorously compared to those of characters from popular culture, especially the television show “Breaking Bad”. This comparison highlights and satirizes the pharmacists’ expertise in chemistry. A comment humorously suggests:


*“What the chemistry professor did in Breaking Bad, a pharmacist in Romania could also do.”*


This allusion to the show implies that pharmacists possess the knowledge and skills to manufacture drugs, albeit in a context far removed from their professional reality.

#### 3.1.4. Leaving the Job Due to Low Salaries and Lack of Satisfaction

The issue of low salaries and resulting dissatisfaction are common themes in these conversations, often addressed with humor. One pharmacist jokes sarcastically:


*“A salary so good you could use your pay slip as toilet paper.”*


Another comment highlights the frustration that drives some to leave the profession altogether for better-paying fields such as IT, using humor to emphasize the seriousness of their dissatisfaction.

*“Well. I gave up and switched to IT. Six months working in a pharmacy was enough for me to throw in the towel* 
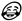
”

His conclusion is consistent with previous research, emphasizing that dissatisfaction and low salaries are the main reasons pharmacists leave their jobs [39,40]. Conversely, factors influencing pharmacists’ decisions to remain in community practice in Ireland include working conditions, career fulfilment, regulatory burdens, commercial pressures, lack of representation, and overall health and well-being, with improvements in these areas likely to enhance retention [41].

#### 3.1.5. Perceived Inferiority Compared to Doctors and Public Recognition

Pharmacists often express feelings of inferiority when compared to doctors, a sentiment frequently articulated through humor. One comment reflects this dynamic:


*“The doctor writes the prescription; you’re just a medicine vendor. I really don’t understand why you need a degree and expect a high salary.”*


This quote expresses the frustration of being viewed simply as someone who provides medications rather than being recognized as a healthcare professional in their own right. Pharmacists often face confrontation because they frequently have to decline various requests from doctors or patients that do not align with the strict guidelines for dispensing and using medication [23]. The communication between doctors and pharmacists is a subject of much research, and different degrees of frustration might appear [42].

Another comment humorously laments and underscores the perceived lack of recognition and respect afforded to pharmacists despite their extensive education and training.


*“You take the money. Place orders and process invoices. Interact with all kinds of crazies and illiterates. The salaries are, at best, mediocre, considering the amount of stress. So no, there’s nothing noble about it. Just a bunch of salespeople with a long and difficult degree.”*


The comparison between doctors and pharmacists is a well-established topic, and some studies focused on pharmacists who transition to medical careers. These pharmacists often do so because they feel their profession is limited and viewed as inferior to doctors; this aspect highlights the perceived disparity in status and recognition between the two professions [43].

#### 3.1.6. Perceived Exposure to Burnout and Occupational Stress

Burnout is another critical issue that pharmacists address with humor. A comment poignantly states,


*“I’ve heard from other pharmacist colleagues that all the local chain owners in Cluj have an agreement not to offer more. We might as well get jobs as supermarket clerks for that money; at least you’d know you’re being paid according to the job’s requirements.”*


His remark highlights the widespread burnout and frustration as pharmacists unfavorably compare their profession to less demanding retail jobs. However, burnout is a complex issue affecting the pharmacist and the organization [4].

Another pharmacist sarcastically remarks,


*“With a line out the door, doing inventory and other nonsense, watching the money to make sure nothing gets stolen from the pharmacy, and with the boss breathing down your neck to tell the sick patient about some raffle, you don’t have time to investigate a prescription, check the diagnosis, and maybe give some recommendations like a true healthcare professional.”*


This comment captures the overwhelming workload and the inability to perform their duties fully, leading to professional dissatisfaction and burnout. A recent analysis shows an estimated 51% to 74.9% worldwide prevalence of burnout among pharmacists [44].

#### 3.1.7. Sarcasm about Financial Transparency and Practices

Humor also addresses concerns about financial practices within the pharmacy industry. One conversation humorously critiques the perceived financial strategies of pharmacy chains:


*“X, we’re getting into details beyond our competence! It’s no coincidence that the law taxes at 2% on revenue all those poor folks claiming zero profit! Even the state doesn’t believe them anymore.”*


This sarcastic comment indicates the scepticism and frustration surrounding certain pharmacy operations’ financial transparency and honesty. Lack of transparency regarding salaries is also mentioned in other regions when discussing salary surveys, mainly due to confidentiality reasons [45].

#### 3.1.8. Using Humor and Sarcasm to Highlight the Overload of Curricula during University Studies

The demanding nature of pharmacy education is another subject of humor and sarcasm. One pharmacist humorously questions the relevance of certain subjects:


*“So many pointless subjects (e.g., Genetics, Immunology). Seriously, why do you need this sci-fi stuff when you end up pushing homoeopathic products to make more money? Forget those who want real medications.”*


This comment reflects the frustration with the extensive and sometimes seemingly irrelevant coursework required during their studies, juxtaposed against the reality of their professional tasks. Another remark:


*“I have three acquaintances in software testing who came from pharmacy. They tried for several years, just like you, until they gave up.”*


This humorously captures the transition from a challenging education to a completely different career, emphasizing the perceived gap between academic preparation and professional fulfilment.

Updating pharmacy university curricula has been a priority for several years, with recent efforts focusing on addressing the challenge of curricular overload [46].

#### 3.1.9. Perceived Redundancy of Pharmacists’ Tasks in the Age of Automation

The public perceives that specific pharmacist tasks may be unnecessary and could be replaced by apps or other automated solutions. For example, a UK pharmacist mentioned an already operational app, leaving the pharmacist with primarily dispensing responsibilities.


*“*
*Google Viagra Connect is (NB—an app) working without a prescription, but you must answer a questionnaire. Then, you receive a ticket that allows you to buy as many as you want for six months without talking to the pharmacist (ideally, you should inform your doctor that you have used Viagra so that he knows).”*


Social media conversations provide valuable cultural insights and humor, offering a multi-dimensional view of the pharmacy profession. They demonstrate how pharmacists use humor to critique their working conditions, express frustrations, and seek solidarity with their peers; this highlights the challenges pharmacists face and emphasizes the importance of humor in fostering a supportive and resilient professional community. Pharmacists navigate the complexities of their profession, finding ways to cope with and reflect on their shared experiences. Some pharmacists’ historical roles may be made redundant, and new roles may be created, decoupling pharmacists to a certain extent from the dispensing and supply process [47].

### 3.2. Economic and Financial Insights

Pharmacists are integral to healthcare delivery and are tasked with responsibilities requiring extensive training and high professional expertise. Despite their crucial role, there is a notable disparity between the compensation pharmacists receive and their significant contributions to healthcare. This study examines pharmacist compensation in Romania by analyzing factors influencing earnings and job satisfaction.

Financial dissatisfaction is a recurring theme that uniquely impacts pharmacists’ economic stability and long-term career satisfaction.

The analysis revealed seven key findings highlighting pharmacists’ economic and financial challenges and these are summarized within Table 3.

#### 3.2.1. Widespread Dissatisfaction with the Level of Salaries

Pharmacists widely report dissatisfaction with their salaries, which do not seem commensurate with the demands of their roles or the profits generated by their workplaces.


*One pharmacist poignantly notes: “I cannot believe that pharmacists in chains receive such miserable salaries… The existence of the pharmacy, its daily operation, and huge profits are achieved through the training and work of the pharmacist. It’s a mockery!”*


This sentiment is echoed across the spectrum, indicating a widespread perception of unfair compensation. The research identifies significant variations in salaries across different regions. Urban centers, characterized by a high density of pharmacists due to local universities and colleges, generally offer lower starting salaries compared to less saturated markets. For instance:


*“Salaries vary greatly depending on the locality. In large cities with universities and many pharmacists, starting salaries are around RON 3200–3500 (EUR 640–700) net.”*


Conversely, in smaller cities with higher demand for pharmacists, opportunities to negotiate better salaries are more prevalent:


*“In smaller cities, with a high need for pharmacists, the salaries are much better… I have even heard of offers over RON 6000 (EUR 1200).”*


It is not surprising that high salaries are linked with greater satisfaction, and we also noticed that salaries are the primary reason for dissatisfaction in other settings [48,49].

#### 3.2.2. Low Entry-Level Salaries and Limited Career Prospects for Recent Graduates

Newly graduated pharmacists often face a harsh reality where their starting salaries do not meet their expectations, impacting their career satisfaction and potential retention in the field:


*“You study for five years to become a mere medicine seller, working on targets, and you will never have a salary higher than RON 4000 (EUR 800) anywhere.”*


Newly graduated pharmacists often struggle to negotiate better terms when entering the workforce. They face challenges such as a lack of confidence, perceived power imbalances with employers, and insufficient negotiation skills preparation, limiting their ability to secure competitive salaries and favorable career prospects [50,51].

#### 3.2.3. Importance and Challenges of Salary Negotiation

Effective negotiation appears crucial in achieving better salary outcomes, though not all pharmacists experience success in this arena. The ability to negotiate can significantly alter individual compensation packages, as illustrated by a pharmacist from Cluj Napoca (the most important university city in Romania and an important medical center):


*“Found a position, negotiated to RON 5000 (EUR 1000), and my surprise, I received several private offers.”*


Another respondent has a different experience with negotiating leverage for a pharmacist. “*Negotiation is an illusion. With 25 years of experience behind me, I have negotiated myself as a regular pharmacist and chief pharmacist. They chose a beginner instead. It’s not about how professional you are; they are more interested in your compliance and minimizing your cost to the company*”.

The findings suggest a need for systemic changes within the pharmacy sector in Romania. Establishing clear salary standards and supporting effective salary negotiation can help align pharmacist compensation with their professional contributions and responsibilities. However, this may be challenging in an open market with limited salary-related boundaries or regulations. The internet is full of articles about how to negotiate salaries for any kind of job, including pharmacists. Most advice is correct, but outcomes depend on market conditions [52].

#### 3.2.4. Job Satisfaction and Compensation

The relationship between job satisfaction and compensation among pharmacists in Romania is also discussed. It focuses on how dissatisfaction with pay rates impacts morale and retention in the pharmacy sector, particularly given the extensive training and critical responsibilities that the profession entails. In pharmacy, where the educational and professional demands are significant, appropriate compensation is pivotal for job satisfaction and professional fulfilment. Many pharmacists report dissatisfaction with their compensation, citing that their salaries do not reflect the extensive training required and the critical nature of their responsibilities. The sentiment of being under-compensated is prevalent, as pharmacists feel their healthcare delivery role is undervalued. A respondent noted:


*“It’s a mockery that the huge profits made by pharmacies are not reflected in pharmacists’ salaries, despite our significant contribution to healthcare.”*


This aspect indicates a general feeling of undervaluation within the profession, contributing to job dissatisfaction, as mentioned in several other research papers [19,53].

#### 3.2.5. Comparison of Salaries with Other Professions

Pharmacists often compare their compensation unfavorably with other professions requiring similar or less educational investment. The disparity is particularly noted when compared with other healthcare professionals or technical roles that offer better compensation for comparable levels of responsibility and training. These comparisons highlight a perceived injustice in the remuneration system, further diminishing job satisfaction among pharmacists. For instance:


*“After five years of intensive study, earning a salary that is on par with or less than professions requiring less training is disheartening. This affects our morale and questions the value of our professional skills in the healthcare market.”*


The findings suggest improving compensation could significantly enhance pharmacists’ job satisfaction and professional engagement. Addressing these compensation disparities is essential for individual morale and maintaining a robust healthcare delivery system that respects and values the contributions of all its professionals.

#### 3.2.6. Career Development Paths

The career path and development opportunities available to pharmacists in Romania in connection with the economic and financial implications give us limited professional growth and the potential benefits of further certifications and specializations. Our findings explore how the evolving role of pharmacists, from healthcare providers to sales-focused positions, affects career satisfaction and financial prospects. The professional trajectory of pharmacists is increasingly influenced by market dynamics and regulatory changes, which can constrain career development opportunities and alter job roles. These were addressed previously in research on the professional satisfaction of pharmacists in Romania [54].

Many pharmacists in Romania express concerns that their professional roles became overly commercialized, focusing more on sales than on patient care. This shift affects their job satisfaction and limits their career development opportunities, as the pharmacist’s traditional role as a healthcare provider becomes marginalized. A pharmacist states:


*“We’ve been reduced to mere salespeople, which diminishes our role as healthcare providers and limits our professional growth within the sector.”*


Such sentiments reflect a broader trend where the core professional identity of pharmacists is at risk, potentially impacting their motivation and long-term career plans.

In contrast to the challenges mentioned above, pursuing additional certifications and specializations, such as residency programs, is viewed as a pathway to potentially better career prospects and higher earnings. These programs are recognized for providing pharmacists with the skills and credentials needed to specialize in clinical pharmacy, pharmacovigilance, or regulatory affairs, thus opening up new avenues for professional advancement and financial improvement. Another respondent elaborates:


*“Engaging in specialized training programs has enhanced my professional capabilities and significantly improved my earning potential, setting me apart in a competitive job market.”*


Such opportunities are crucial for pharmacists who seek to transcend the limitations of the prevailing sales-oriented job roles. While commercial pressures may hinder traditional roles, specialization and advanced certifications offer routes to reclaim professional stature and achieve financial gains. Addressing these challenges requires a balanced approach that includes policy support for continuous professional development and recognition of specialized skills in compensation structures.

#### 3.2.7. Market Dynamics and Indication of Salary Levels

The interplay between market dynamics, employment opportunities, and financial stability within the pharmacy sector in Romania influences the economic and financial aspects of pharmacist life. Urban market saturation affects wages, the potential benefits of employment in less saturated markets or specialized roles, and the profession’s overall financial stability and growth prospects. The pharmacy sector in Romania faces unique challenges and opportunities shaped by market dynamics, economic conditions, and healthcare policies.

In urban centers, the abundance of pharmacy graduates led to an oversupply of pharmacists, which depresses wages.


*One respondent notes, “In large cities, the competition among pharmacists is fierce, resulting in lower starting salaries and limited wage growth.”*


Another respondent, referring to differences in salaries, said,


*“Salaries vary greatly depending on the location. I know that in large cities with universities and many pharmacists, starting salaries are around RON 3200–3500 (EUR 640–700) net. In smaller cities, where there is a greater need for pharmacists, the salaries are much better…*
*”*


Salary ranges are also important. Based on the analysis of these conversations, we can say that net salaries for pharmacists vary between RON 2300 and 10,000 (EUR 460–2000). Please note that EUR 1 equals RON 5, and the average salary in Romania is RON 3700 (EUR 740). Salaries for entry-level pharmacists generally start around RON 2300 (EUR 640) and can go up to RON 3500 (EUR 700) in urban areas with higher saturation.


*“I graduated in 2020: my first job offered a salary of around RON 2200 (EUR 440), plus various targets. I was also deceived.”—highlighting the struggle with low entry-level wages.*



*“In large or university cities, salaries will always be lower because the supply of pharmacists exceeds demand.”—indicating the impact of market saturation on starting salaries.*


In less saturated markets or specialized roles, salaries can range from RON 4000 (EUR 800) to over RON 6000 (EUR 1200), with opportunities reaching RON 8000 (EUR 1600), including benefits from residencies or specialized positions. High-end salaries in less competitive markets, for pharmacists with significant experience, or in managerial roles can exceed RON 10,000 (EUR 2000).


*“Salaries vary greatly depending on the location. I know that in large cities with universities and many pharmacists, starting salaries are around RON 3200–3500 (EUR 640–700) net. In smaller cities with a high demand for pharmacists, salaries are much better… I’ve even heard of offers exceeding RON 6000 (EUR 1200).”—demonstrates the variation based on geographic location.*



*“In the industry, you have production, quality assurance (involving numerous documents and procedures to remember and implement), quality control (you need to be proficient in laboratory practices including chemistry, spectrometry, HPLC, and GC, and possess steady hands and attention to detail), and regulatory affairs (where it is crucial to have a thorough understanding of the laws in other countries and know what documents are necessary for product authorization, etc.).”—pointing to specialized roles that often offer higher compensation.*



*“As an example from the pharmaceutical industry, I had a salary of about RON 5000 (EUR 1000), and I earned an additional RON 3000 (EUR 600) from my residency, totalling around RON 8000 (EUR 1600).”—illustrates the financial benefits of combining professional work with residency programs.*


These quotes and salary data underscore the range of compensation pharmacists can expect in Romania. The variability is primarily influenced by factors such as geographic location, market saturation, the specific nature of the employment setting, and the level of specialization. Addressing these disparities and ensuring fair compensation will be crucial for enhancing job satisfaction and retaining skilled professionals within the pharmacy sector.

Salaries may be nominally higher in urban areas such as Cluj or Bucharest, but the cost of living significantly reduces the real income.


*“If you are paid over 5000 (EUR 1000) in Suceava, with the costs you have in Cluj, you will need a care package from home!”*


Some pharmacists consider moving to regions with a better salary-to-cost-of-living ratio to enhance financial well-being.


*“I’m curious: What do you consider a miserable salary? As a mechanical engineer, salaries start from RON 3000–3500 (EUR 600–700) net and barely exceed 5000 (EUR 1000) after a few years (equally miserable). It seems that emigration remains the solution.”*


There is a call for policies to ensure compensation that accounts for the cost of living, suggesting that there should be a minimum wage for pharmacists in other Western countries.


*“Why don’t you rather advocate for a minimum pharmacist salary imposed on employers, like in the West?”*


This analysis confirms that the geographic location significantly affects the financial stability of pharmacists in Romania, with salary disparities relative to the cost of living between urban and rural settings. Addressing these issues involves policy intervention, education on financial negotiation, and perhaps geographic redistribution of pharmacy services.

### 3.3. Global Trends and Technological Impacts

Social media discussions among pharmacists provide insights into global trends and technological impacts that affect their profession. This category explores international pharmacy practices and the implications of AI, revealing diverse perspectives and challenges. The pharmacy sector is experiencing a profound digital transformation, with technological advancements such as artificial intelligence, blockchain, and online platforms significantly reshaping pharmacy services, education, and global practices. This shift underscores the evolving landscape of the profession in response to rapid technological changes [55].

Analyzing the snippets within this category produced three key findings that highlight comparative practices, global mobility, and technological impacts on pharmacy presented in Table 4.

#### 3.3.1. Appealing Pharmacy Models (Hungary and USA)

Pharmacists in Hungary and the USA benefit from different models that provide higher salaries and better working conditions than those in Romania. The Hungarian model is particularly praised for the respect and professional recognition pharmacists receive and their substantial salaries.

*“The Hungarian model is very effective! The country’s leadership has resolved the issues there… Pharmacists are given the title of ‘Doctor’, and they are respected and well-paid* (Highlighting the respect and better salaries in Hungary)

In the USA, large pharmacy chains offer high salaries, and unions play a significant role in negotiating these wages. However, these models also highlight the current system’s shortcomings in Romania, where pharmacists feel undervalued and underpaid. There is a strong call for establishing minimum wage standards and the role of unions in salary negotiations to improve the situation.

*“Why not advocate for a minimum wage for pharmacists imposed on employers, like in Western countries?”* (Calling for a minimum wage for pharmacists in Romania)

*“In the USA, salaries are negotiated by unions and are standardized. For example, an entry-level pharmacist earns around USD 25 per hour.”* (Emphasizing the role of unions in salary negotiations in the USA).

These statements are beliefs held by participants in various conversations and represent models that other members of different social media groups consider as the basis for discussions. It is not clear what these models look like or in what ways they are favorable to pharmacists.

#### 3.3.2. Global Mobility for Pharmacists (UK and Ireland)

Pharmacists can move and work in other countries, primarily requiring them to pass a language exam. The UK and Ireland are notable examples where pharmacists can achieve higher salaries and better working conditions. These opportunities provide a pathway for pharmacists seeking better professional and financial rewards, emphasizing the importance of language proficiency and the recognition of foreign qualifications.

*“My friend graduated from the University of Timișoara and worked for a few months with a net salary of RON 2300 (EUR 460). Then, we both moved to the UK. After credential recognition and passing the IELTS, a pharmacist’s salary starts at GBP 32,000 in the West Midlands.”* (Highlighting the benefits of moving to the UK).

Pharmacy models in Hungary and the USA are seen as more appealing, while global mobility offers opportunities in countries such as the UK and Ireland.

#### 3.3.3. Significant Impact of AI on Pharmacy Work

AI and automation are anticipated to impact the work of pharmacists significantly. Many tasks currently performed by pharmacists, such as medication counselling and interaction verification, could be replaced by AI, raising concerns about job redundancy. While AI can perform these tasks more accurately and efficiently, there remains a strong preference for human interaction in pharmacy services; this highlights the potential need for pharmacists to upskill and adapt to the changing technological landscape to stay relevant in the profession.

*“I don’t think there will be many pharmacists in the future… an automated dispenser can easily replace a pharmacist. All the checks that a pharmacist performs can be done electronically with greater accuracy.”* (Expressing concerns about AI replacing pharmacists)

Social media analysis of pharmacists reveals global trends and technological advancements impacting the profession. AI and automation present opportunities and challenges, requiring continuous professional development and upskilling. Addressing these areas will ensure pharmacists are valued and fairly compensated for their vital contributions to healthcare [56,57].

### 3.4. Personal Experiences and Satisfaction

The satisfaction and personal experiences of pharmacists offer a clear understanding of the realities of the profession. Through social media discussions, pharmacists shared their insights on various aspects affecting their daily lives and professional fulfilment; this category covers a wide range of experiences, highlighting the challenges and opportunities that shape pharmacists’ career paths and job satisfaction. The analysis of these discussions produced key findings that reflect the nuanced dynamics of working within the pharmacy profession.

Table 5 summarizes key findings in the “Personal Experiences and Satisfaction” category from social media discussions among pharmacists.

#### 3.4.1. Working in Profit-Oriented Chains and Greedy Managers

The profit-driven nature of pharmacy chains, which usually results in inflated performance expectations and unethical behavior, is a common source of frustration among pharmacists. The manipulation of finances to lower apparent earnings is a recurring problem impacting pharmacists’ pay and job security. For instance, one pharmacist noted that in their chain,


*“The profit of the pharmacy chain where I work was just over [redacted] in 2022. At least, that’s what was reported. In the past, I worked at another chain where, in the last accounting month of the year, spaces or other assets were purchased to show lower profits,”*


indicating practices designed to minimize declared profits.

Furthermore, the hiring process is frequently dishonest, with managers portraying the highest possible bar during interviews to entice candidates—only for these barest to vanish after the candidate is employed:


*“Plus, during the interview, they always highlight the maximum target to attract you. Likewise, if you have to work weekends or extra hours, make sure that the payment for these is not included in the same RON 3900 (EUR 780).”*


Such practices lead to frustration and mistrust among pharmacists.

Previously cited papers confirm that pharmacists working with pharmacy chains evaluated professional satisfaction at a lower level, although the level of services delivered should be similar [48,58]. A survey conducted among hospital pharmacists in Canada found that making a difference and being recognized were crucial elements in driving professional satisfaction, which can be challenging to attain in a commercial environment [59].

#### 3.4.2. Low Salaries and Poor Working Conditions

The prevalent issue of low salaries, despite extensive education and experience, is a significant source of discontent. Many pharmacists report insufficient earnings, especially considering their qualifications and responsibilities. One comment highlights this disparity:


*“After five years of studying, I barely earn RON 4000 (EUR 800) per month.”*


This comment reflects on the inadequate compensation relative to the years of study and effort invested.

Unfavorable working circumstances, such as excessive hours and weekend jobs without proper pay, also worsen this discontent. Pharmacists often feel undervalued and overworked, with one stating,


*“In pharmacy chains, employees are treated like cannon fodder,”*


emphasizing the lack of recognition and respect in their roles.

As previously mentioned, workload and low pay are cited as the most critical factors causing dissatisfaction among pharmacists [2,10]. Our findings partially confirm the results of a recent study in Romania on the sources of professional dissatisfaction, particularly the lack of proper pay [60].

#### 3.4.3. Challenging Work Environment and Stressful Interactions

A relative of a pharmacist with 18 years of experience shared a particularly disheartening perspective. She worked in both large chain pharmacies and a small pharmacy in Bucharest, so she does not recommend the profession to anyone. Daily, she encounters customers demanding prescription medications without proper prescriptions and individuals seeking diagnoses for their ailments, creating a constant atmosphere of conflict. Additionally, she often deals with people requesting substances typically used for drug creation, who can become violent when refused. This stressful environment significantly impacts job satisfaction and contributes to the reluctance among seasoned pharmacists to recommend this career path to new entrants.


*“(…) She has been a pharmacist for 18 years. She worked at large pharmacies in our country and now works at a small pharmacy in Bucharest. She told me she would not recommend anyone to become a pharmacist. Every day, people come in wanting prescription medications without proper prescription, and if they don’t get them, they cause a scene. (…). Every day, suspicious individuals come in requesting medications that are commonly used to make drugs, and if she refuses, some become violent.”*


A survey of pharmacists in the Arab world highlighted underestimation of their role, low salaries, lack of motivation, and excessive workload as significant factors contributing to job dissatisfaction, emphasizing the challenging work environments and stressful interactions faced by pharmacists in these regions [61].

#### 3.4.4. Better Opportunities in Alternative Jobs

Many pharmacists consider or ultimately transition to alternative careers that offer better pay and conditions. Positions in industry, clinical research, or even unrelated fields such as IT are viewed as more rewarding. For example, one pharmacist noted,


*“I switched to IT. It’s incredibly frustrating. Five years of college, all those technical subjects (chemistry, thermodynamics, physics, etc.), only to end up either as a salesperson with criminal liabilities (as a pharmacy manager or pharmacist in general) or in sales, where you don’t need all that schooling and chemistry. The salaries in pharma aren’t even worth mentioning. I lasted six months in a pharmacy and a few more in the audit field, and I was done. I gave up.”*


This highlights the trend of pharmacists moving to IT for better opportunities.

These findings are consistent with previous research results, which indicate that 44% of New Zealand pharmacy graduates plan to shift their profession within five years [62].

#### 3.4.5. Young Pharmacists and Career Mobility

Younger pharmacists in particular, are quick to seek better opportunities, often leaving the profession shortly after gaining some experience. This mobility is partly driven by realizing they can secure better pay and conditions elsewhere. A poignant example is a recent graduate who felt disillusioned:


*“I recently graduated from pharmacy school and got a job in a community pharmacy. The salaries are very low, the working hours are long and hard, and my team is not the friendliest. I work 10 h days on weekends, one weekend every two weeks, and during the week, I work 8 h days.”*


#### 3.4.6. Salary Disparities among Pharmacists

The disparity in salaries between new graduates and experienced pharmacists is another significant issue. New graduates often receive similar or higher salaries than their more experienced counterparts, leading to dissatisfaction and resentment. One pharmacist mentioned,


*“The salaries in this field are terrible, but it’s not fair for a beginner pharmacist who hasn’t faced practical situations to earn the same salary as someone with 10–20 years of experience.”*


Our findings contrast with a U.S. study, which highlighted wage inequalities across various demographic and job-related groups, as we observed significant salary disparities where new graduates often earn similar or higher salaries than more experienced pharmacists, leading to dissatisfaction and resentment [63].

#### 3.4.7. Migration for Better Opportunities

Pharmacists frequently consider moving abroad for better pay and working conditions. Countries with higher standards for pharmacists are attractive options, offering financial and professional respect that they feel is lacking in their home country. One pharmacist’s experience in Ireland illustrates this:


*“I don’t know much about the field or your opportunities in Romania. What I can say is that I have a close friend who was in a similar situation, disappointed with the pharmaceutical industry in Romania and the prices. She found a job as a pharmacist in Ireland and is doing very well. She supported her husband and child on her salary alone for one year, as he couldn’t find a job. This is something to consider if you are thinking about changing countries.”*


Migration opportunities are a key factor for Romanian pharmacists, as shown by the high number of Current Professional Certificate requests. There were peaks in 2015 and 2016, followed by a decrease in 2017 and 2018, indicating a trend of seeking better opportunities abroad, according to 2022 research [64]. Similarly, due to economic reasons and wage disparities, India experiences the same trend in pharmacist migration [65].

#### 3.4.8. Challenges in Academia and Research

Working in academia or research is also fraught with challenges. The environment can be highly competitive, with limited advancement opportunities and financial rewards. As one former academic noted,


*“I left university teaching after 15 years with a PhD. At that time, the salaries were miserable. The salaries are reasonable now, but if we don’t have students… the future is uncertain.”*


Although the potential lack of new students is a recent finding, the low pay and politics are cited as unattractive factors for academia in other studies [66].

#### 3.4.9. Passion and Fulfilment in Work

Pharmacists must work with passion if they hope to succeed professionally; this frequently entails helping them choose careers that fit their values and areas of interest. Despite the difficulties, some chemists stay in the industry because they have a strong passion for it. One pharmacist expressed this dedication:


*“I am not leaving because this is what I studied. I love this profession and do not want to give free rein to the chains!”*


This topic is the subject of research in Australia involving pharmacists in rural outlets, and our findings are consistent with that [67].

#### 3.4.10. Career Options beyond Community Pharmacies

Pharmacists dissatisfied with community pharmacy work can explore other areas, such as industry, clinical research, or regulatory affairs. These roles often offer more rewarding experiences and better compensation. One pharmacist advised,


*“If you are unhappy in community pharmacies, you can move to other fields such as industry or clinical research.”*


This pharmacy diploma opens up various opportunities, including roles in community pharmacies, the pharmaceutical industry, sales and marketing, clinical trials, and academia and research [41,68].

#### 3.4.11. Choosing Pharmacy for Interest in Chemistry

Many students choose pharmacy because of their interest in chemistry, only to find the reality of the profession differs from their expectations; this can lead to disillusionment and a desire to change career paths. One student noted,


*“I chose pharmacy because I liked chemistry and all those stories, but what you do in university doesn’t really match what you dreamed you would be doing.”*


#### 3.4.12. Difficulties for New Graduates

New pharmacy graduates often face tough times when starting their careers. They encounter a gap between their education and the practical demands of the job, leading to a steep learning curve and additional stress. One recent graduate shared,


*“I graduated five years ago and have been working since my fourth year, and no, my responsibilities were not just those of an operator.”*


Analyzing pharmacists’ discussions reveals deep unhappiness within the profession. Low pay, unfavorable working conditions, and management focused on profits are responsible for this. Many pharmacists think about quitting their jobs in search of better prospects in other industries, different countries, or even different capacities within the pharmaceutical sector.

### 3.5. Professional Growth and Education

The professional growth and education field for pharmacists covers many experiences, opportunities, and challenges and it is summarized in Table 6. From navigating challenging university curricula to exploring different career paths in industry, academia, and abroad, pharmacists encounter many factors influencing their professional paths. This section examines these aspects based on data and feedback from professionals in the field.

#### 3.5.1. Difficult University Curriculum

Pharmacy students often face a challenging university curriculum, with some subjects perceived as irrelevant to their future careers. The initial years in particular, are criticized for being filled with content that does not directly contribute to their roles as pharmacists. This sentiment is captured in a student’s comment:


*“For me, the first two years were filler; I didn’t learn much, many subjects were pointless…”*


The student highlighted the perception that initial years include subjects that do not directly contribute to their future roles as pharmacists.

Additionally, there is a need for skills beyond what is taught in university, as highlighted by a comment:


*“(…) you need skills that no one teaches you in faculty, such as financial education, marketing, business, negotiation, in addition to medical-related knowledge, staying up-to-date with new developments, providing quality services, etc. Yes, it’s harder now than in the early 2000s when a dinosaur could remain with the same knowledge for years.”*


Several studies we came across raised concerns about the need to update pharmacy curricula [69,70,71,72,73]. However, we never found any information about initiatives to update pharmacy curricula in Romania.

#### 3.5.2. Perceptions of Pharmacy Universities

The declining interest in Romanian pharmacy schools is evident, with fewer applicants due to perceived difficulties and low financial rewards. One comment underscores this trend:


*“This year in Iași, only 45 students applied for admission for the 90 available tuition-free spots.”*


The comment pointed to decreasing the level of applications due to perceived difficulties and low professional financial rewards. These perceptions contribute to the reduced competition for admission to pharmacy universities, signalling a need for reforms to attract and retain talented students.

#### 3.5.3. Residency Programs for Better Income

Specialization through residency programs is pursued not only for professional development, but also for financial benefits. Residency programs provide additional income while allowing pharmacists to balance multiple roles. One comment states:


*“Yes, residency specialization is worth it, as you get about RON 3500 (EUR 700) during three years of training, and you mostly make online courses,”*


indicating that these programs provide a steady income and the opportunity to work additional jobs simultaneously.

Our scoping review suggests that specialized pharmacists could expand their current roles, potentially improving both patient care and the efficiency of healthcare systems [74].

#### 3.5.4. Career Growth in Industry and Authorities

The pharmaceutical industry and regulatory bodies are considered promising career advancement areas. These sectors offer opportunities for pharmacists to move from entry-level positions to management roles. A professional recommends:


*“I recommend creating a LinkedIn account and following all pharmaceutical companies and CROs in Romania…”*


The pharmacist suggested active engagement on professional networks to explore opportunities in these sectors.

These insights suggest that active engagement in professional networks can significantly enhance career prospects in the pharmaceutical industry and regulatory authorities, and these alternative job options are regarded better than those of community pharmacies.

#### 3.5.5. Going Abroad for Better Prospects

Many pharmacists consider relocating abroad for better salaries and career growth opportunities. The financial and professional benefits of working in other countries are substantial. An example provided by a pharmacist highlights this:


*“Equivalency (of pharmacist’ diploma) + IELTS, a pharmacist’s salary starts from GBP 32,000 in West Midlands…”*


The comment illustrated the significantly higher pay and improved conditions in other countries. The interest of Romanian pharmacists in working abroad was already discussed in the Section 3.4.7.

Overall, pharmacists’ professional growth and education landscape is marked by a complex interplay of educational challenges, strategic career moves, and the pursuit of better opportunities both within and outside the country. Addressing these challenges requires a concerted effort from academic institutions, industry stakeholders, and regulatory bodies to create a more supportive and rewarding environment for current and future pharmacists.

### 3.6. Professional Identity and Public Perception

The subject of “Professional Identity and Public Perception” focuses on how the general public and the healthcare industry view pharmacists, considering preconceptions, stereotypes, and the degree of professional recognition they receive.

Examining the excerpts in this category led to the following eight key findings that emphasize pharmacists’ perceptions in the medical community and in general, including the degree of professional recognition, misconceptions, and stereotypes (Table 7).

#### 3.6.1. Changing the Attitude of Pharmacists Related to the Professional Organization

Changing the public’s attitude toward pharmacists is essential in redefining their role. If the representative organization is absent, pharmacists have to refuse low salaries similar to unskilled workers for many hours at weekends.

*“Learn to say NO. If the Romanian Chamber of Pharmacists* (NB professional association of pharmacists in Romania) *do not represent us properly, we must set limits at an individual level that we do not cross. Stop accepting unqualified salaries and 10–12 h shifts on weekends; the change will happen eventually, regardless of the risks. You will not be satisfied if, after a month with two weekends worked, you earn an income of about 5000 (EUR 1000); in fact, the real income is less than 4000 (EUR 800), and the rest is overtime.”*

The lack of clear regulations regarding pharmacists’ remuneration and the expected low or no additional salary pay for pharmaceutical services amplifies dissatisfaction with the professional organization. It is felt that the organization does not adequately represent pharmacists and does not fight for their rights.


*“You are an independent profession, and you set some rules. Of course, by law, you can equalise pharmacists’ salaries in the private sector with those in the hospital or ask that part of the employer’s salary burden be taken over by the state. As long as you have a monopoly on the price of RX (prescriptions), you only have to pay the pharmacist for prescription processing and counselling. But for this, we need representation. Don’t you understand that it is not wanted? Not that it can’t be done.”*


The affiliation with a professional organization is intended to provide professional development opportunities and education and outline financial obligations towards such an organization. Additionally, the requirement for membership as a pharmacist to practice is among the topics discussed in previous research [75]. Implementing new pharmaceutical services and an extended role within healthcare for pharmacists requires effort from professional bodies [76].

#### 3.6.2. Disappointment about Low Salaries Compared to Other Professions

Many comments criticize the uncompetitive salaries of pharmacists. The inequity between doctors and pharmacists is also targeted. Numerous comparisons are mentioned both with other unskilled and skilled jobs; for example, helpdesk, hypermarket saleswoman (Cora), market cashier (Lidl), city councilor, manicurist, IT-ist, customer support, ice cream seller, grocery store cashier, and nurse. Some of the answers are reproduced below.


*“The salary of a pharmacist is equal to that of a saleswoman in Cora. It can’t be like that anymore.”*


Legal regulations and remuneration for pharmaceutical care are crucial for successful implementation in Poland, according to research performed in 2021. Authorities and health insurance companies should understand the clinical and economic effectiveness of pharmaceutical care services, establish a reimbursement system, and address financial aspects [77].

#### 3.6.3. Misconceptions: Pharmacists Are Salespeople, Not Healthcare Providers

There are wrong perceptions outside the profession, and some stereotypes exist regarding the pharmacist and pharmaceutical fields. Several discussions focus on the fact that pharmacists have a commercial-driven target, which is pressure from employers. Thus, pharmacists are forced to promote various products (e.g., dietary supplements and homoeopathic medicines) that are considered ineffective.

The institution of “pharmacy” was reduced to the trade of medicines and the pharmacist to a drug seller. Some comments say the pharmacist can be replaced with a medicine dispenser anytime.


*“I fail to see the necessity for them anymore. Perhaps having a person at the counter is beneficial only for those who cannot read. Otherwise, it is feasible to substitute the pharmacist with an automated dispenser.”*


Unfortunately, pharmacists are considered to be poorly trained professionally. In addition, they are frequently confused with pharmacy assistants, who are permanently in the office.


*“Those who sell medicines most often completed post-secondary school as a pharmacist assistant.”*


This statement is a very painful misconception for pharmacists, and mainstream media take these stories out for the general public, underlying the perception of a pharmacist as a salesman [78].

Another wrong perception is that today’s pharmacies no longer prepare medicines. In the comments, a critical response appeared from a pharmacist who works in a pharmacy that prepares medicines.


*“I work in a pharmacy outside the country, and here it is a requirement that every pharmacy has a laboratory where, surprise (!), medicines are prepared. Some dosages are not found in the industry, such as capsules for children with heart conditions.”*


Pharmacists are dismayed by their limited recognition of being key healthcare providers within the system.


*“You should know that the pharmacist profession extends beyond the community pharmacy, where in most cases, yes, it is as you said for certain reasons, but from there, until you come here to denigrate an entire profession without having an overall picture formed, it seems a bit exaggerated to me.”*


From the analysis of the conversations, it appears that the faculty of pharmacy is no longer desirable because pharmacists are generally perceived as drug sellers.


*“What our universities produce now are not pharmacists but drug sellers; their only quality is to be able to read the doctor’s writing. …Now, I’m sorry to tell you, but you are glorified drug salesmen and nothing you have or have not learned is applicable.”*


Due to chain pharmacies being inherently commercial, many patients have a poor impression of pharmacists and see them more as salespeople than as medical specialists. Pharmacists face ethical dilemmas when they confront pressure to promote medications or services that may not align with the best interests of their patients due to the focus on sales and profitability.

The 2018 review uncovered a notable disparity between pharmacists’ true duties in selling complementary medicines, such as providing guidance and ensuring safe usage, and the widespread misconception of them as mere salespeople. This misunderstanding is worsened by the absence of a coherent ethical framework governing pharmacists’ responsibilities, underscoring the necessity for guidelines that emphasize their role as healthcare providers rather than just sellers [79].

#### 3.6.4. Frustrations about the Lack of Recognition of the Profession’s Status

Several factors contribute to the lack of recognition of the pharmacist profession, such as low salaries, doctors’ superiority, the professional organization’s indifference regarding the rights and problems faced by pharmacists, and pressure from employers to increase sales by any means.


*“If you want to be frustrated forever because you know more than everyone else about medicine, but you are continuously humiliated by doctors, by those who own the pharmacies, by the college of pharmacists, and by Romania (the country) in general, become a pharmacist.”*


Pharmacists face a complex ethical landscape where their role as healthcare providers intersects with broader social and political issues affecting patient health, yet guidance on navigating these domains is lacking. This complexity, coupled with external pressures to prioritize sales and other factors undermining professional recognition, contributes to a disconnect between pharmacists’ actual responsibilities and public perception of their role [80]. Interprofessional collaboration is fundamental to improving patient care, especially in managing chronic diseases such as cardiovascular conditions [81]. According to a survey performed in Pakistan, pharmacists were willing to perform their duties and provide healthcare benefits to patients. However, they seemed skeptical of advanced clinical roles, highlighting the need to increase awareness of pharmacists’ significant healthcare contributions [82].

#### 3.6.5. Pharmacists’ Concerns: Corporate Takeover Affects the Professional Image

According to many comments identified in our study, the expansion of pharmaceutical chains contributed to the deterioration of the pharmacist’s image to a large extent. The pharmacy regulations in Romania allow for multiple pharmacy owners, so a pharmacy does not need to be owned by a pharmacist. Pharmacy chains dominated the market in the last 10–15 years. Independent pharmacists are struggling to survive due to unfair competition from big pharmacy chains with superior access to financial and marketing resources.


*“In the old days (not too far away), people also flocked to the pharmacy faculties (also difficult, by the way) because the pharmacies were independent and you, as a pharmacist, practically owned your own business. Over time, the market was gradually taken over by the large pharmaceutical chains that expanded strongly, opened pharmacies from 2 by 2 m and took over the traditional pharmacies piece by piece, having a completely different influence (legislative lobby, non-stop advertising on TV, high incomes, tax optimization, etc.).”*


Pharmacists working in chain pharmacies encounter several difficulties that may affect their level of job satisfaction and the standard of treatment they offer (e.g., high workload, time pressure, staffing shortages, additional customer service tasks, and less autonomy than independent pharmacy employees). Due to chain pharmacies’ commercial orientation, the public may see pharmacists less as health practitioners and more as salespeople.

This issue is a public theme, and almost every week, the national press publishes stories about the pressure that greedy corporations put on pharmacists in the USA. An example can be found in a recent NBC News article discussing how pharmacies are understaffed in pursuit of profit targets and creating an unfriendly environment for pharmacists [83].

#### 3.6.6. The Pharmacy Profession Is Widely Respected

Opportunities in other pharmaceutical specializations (e.g., clinical studies, regulatory affairs) bring professional satisfaction to pharmacists willing to work outside community pharmacies and obtain professional satisfaction. From collected conversations, these specialities seem less known than the job of a pharmacist in a community pharmacy. Some comments emphasize that the public’s perception of a pharmacist is much better than the pharmacist’s view of the profession.


*“I know someone who finished faculty 6–7 years ago, did his residency and then got a position as a clinical trial monitor at a large pharmaceutical company. He worked hard and seriously and earned very well. He tells me that they are always looking for competent and serious people.”*


Other sources confirm that the public regards pharmacists highly in terms of ethical standards and honesty [84].

### 3.7. Regulatory and Market Environment

The regulatory and market environment significantly influences the pharmacy profession, shaping operational practices and economic outcomes. Discussions among pharmacists highlight the challenges and opportunities within the regulatory framework, ownership models, labor regulations, and market dynamics. This section delves into six key findings (Table 8) reflecting pharmacists’ sentiments and experiences navigating these complex issues.

#### 3.7.1. Ownership and Market Consolidation

Romania’s pharmacy industry is highly centralized, with large chains controlling the market and putting independent pharmacies under pressure. Pharmacists are looking for sole ownership arrangements to increase profitability and control. Independent pharmacies are disadvantaged due to the complexity of chains’ vertical integration.


*“(.) chains have their warehouses. They get medicines at huge discounts. For independent pharmacies, medicines barely arrive and don’t get them. So, where is the profit and high salary supposed to come from? There may come a time when we wish for pharmacies with ‘independent owners’… but it will be too late… the chains will dictate salaries and high prices….”*



*“Who are you going to fight? Seven chains? It’s a business; this isn’t going to change in this context; no one will remove them from where they’ve settled. Who will change the rules? In chains, employees are cannon fodder. I have a colleague who became ill after more than 20 years in chain X and another colleague who died from stress in the magnificent chain Y. I also know plenty of people who left as far as they could when they saw they were being destroyed and were just numbers meant to generate margin.”*


According to a study conducted in South Africa or Canada, multiple ownership models align with business objectives, but do not improve access to medicines [85,86].

#### 3.7.2. Regulations on Pharmacist Presence

Regulatory requirements for pharmacist presence in pharmacies are a point of contention. There is a call for differentiated regulations based on the size of the pharmacy. Suggestions include additional compensation for chief pharmacists who bear significant responsibility. The current uniform regulations are seen as inadequate and burdensome for smaller pharmacies.


*“(…) The hyperpharmacy with 2000 square meters is authorized by a single pharmacist, just like the one with a minimum of 50 square meters. The rest of the employees don’t need pharmacy university education. This aspect is all to the detriment of the profession.”*



*“The (chief) pharmacist’s license has to be paid for!!! Stop fooling pharmacists!!! Stop making a mockery of them with miserable salaries, especially in university centres!!!”*


#### 3.7.3. Taxes and Economic Regulations

Economic laws and hefty salary taxes significantly impact pharmacies’ ability to remain financially viable and pay higher salaries. The current business model is putting pressure on community pharmacies current business models, making it more challenging to pay pharmacists competitive salaries. Economic reforms must be implemented to overcome these issues.


*“Do you know that a net salary of RON 5000 (EUR 1000) means a gross salary of approximately RON 10,000 (EUR 2000)? Calculate how many medicines, OTC products, etc., must be sold to achieve this amount from the markups.”*



*“We must fight for our rights: higher markups, lower taxes, (real) recognition for our professional education!”*


#### 3.7.4. Labor Regulations and Professional Pressures

The perceived lack of clarity in labor legislation surrounding work hours, particularly on weekends and extra hours, places a great deal of strain on professionals. They are under more stress and commercial pressure because they perceive the professional organization (Chamber of Pharmacists—CFR) as ineffectual in speaking out for pharmacists.


*“Learn to say NO. If the CFR does not represent us adequately, we need to set individual limits that we should not exceed.”*


#### 3.7.5. Market Realities and Pharmacist Availability

Pharmacist availability varies considerably. Although they frequently offer good incomes, small communities have trouble drawing in professionals. However, there is an overabundance of pharmacists in university cities, which causes big pharmacy chains to pay their employees poor wages. These market factors bring about disparities in work prospects and circumstances.


*“For months and years, I have been looking for a pharmacist on a salary scale higher than RON 4000 net… I haven’t been able to lure even one person out of the chains everyone complains about.”*


#### 3.7.6. Practice Standards and Commercial Focus

The pharmacy profession is seen by many as being too focused on commercial interests, which took attention away from its primary healthcare duties. There are calls for more explicit practice standards and potential government support to shift the emphasis to professional and patient-centered care. This aspect includes suggestions for partial government funding to reduce the commercial pressures on pharmacist salaries.


*“You are not ethical with the patient; you just aren’t because you have targets to meet, and it’s impossible not to push something onto them.”*


Romania’s market and regulatory environment for pharmacists includes several difficulties, such as ownership structures, tax laws, labor laws, and professional standards. To preserve a sustainable and fulfilling professional landscape, legislators, professional associations, and pharmacists are called to work together to address these concerns.

### 3.8. Workplace Dynamics

The “Workplace Dynamics” category emerged as a significant theme in our qualitative research on pharmacy professional satisfaction based on pharmacists’ conversations on social media. This analysis delves into the intricacies of workplace dynamics, focusing on the relationship between employees and management, the working conditions, and the overall environment within pharmacy settings (Table 9).

#### 3.8.1. Difficult Working Conditions

Pharmacists often find themselves dealing with challenging working conditions. These conditions are characterized by hectic schedules, long working hours, and the need to work on weekends. On top of this, they face considerable pressure from both their management and the patients they serve.


*“With a line out the door, managing inventory and other trivial tasks, keeping track of cash to prevent theft from the pharmacy, and with the manager constantly breathing down your neck to push patients into some pointless raffle, there is no time to properly review prescriptions, understand the diagnosis, and provide professional recommendations as a true healthcare professional should*
*.”*


Many pharmacists emphasize the difficulties of balancing the pressure to meet sales targets with their patients’ ongoing and varied needs. They often find themselves in a situation where their efforts are not adequately compensated.


*“It’s a joke: low salaries (you earn the same as if you worked at McDonald’s), an unbearable schedule (there are pharmacies in shopping malls with 10 h shifts on Saturdays and Sundays with only one pharmacist on duty, which is you), plus you won’t have time to be a pharmacist, you’ll be strictly a vendor. And you don’t work with just any clients, but with sick patients who will mentally exhaust you. There is an ongoing medication shortage, and patients’ dissatisfaction will be directed at you.”*


Furthermore, there is apprehension about potential breaches of legal regulations concerning weekly rest days.


*“So I work Monday to Friday, 8 h a day, then have one weekend off. The following week, I work Monday to Friday, 8 h a day, and then on Saturday and Sunday, I work 10 h each day. After that, I will start again on Monday morning, but next weekend is off. Essentially, I work every other weekend.*
*”*


#### 3.8.2. Better Working Conditions in Smaller Towns

Pharmacists working in smaller towns often experience more favorable working conditions, including higher salaries, fewer weekend shifts, and greater schedule flexibility. However, many pharmacists are reluctant to relocate or commute to these areas despite the better terms:


*“I know a pharmacy 5 min from (redacted, big city) that is looking for a pharmacist and offers a salary that reaches RON 6000 (EUR 1200) (net) with one weekend worked, plus meal vouchers. And no one wants it, so…”*


#### 3.8.3. Opportunities in the Pharmaceutical Industry

Many pharmacists find better working conditions and higher salaries in the pharmaceutical industry, particularly in regulatory affairs, quality control, clinical trials, and sales and marketing roles. Bucharest and Targu Mures are noted as significant hubs for pharmaceutical manufacturing:


*“You can also work in factories. The highest concentration of pharmaceutical factories is in Bucharest and Mures. Salaries are okay, similar to community pharmacies, but the schedule is much more regular with weekends off. In pharma sales, pharmacists are not often hired anymore. However, if you have strong industry skills, there are opportunities in regulatory affairs, vaccines, market access, and domain lead roles. And if you know research, there are plenty of jobs in clinical research.*
*”*


#### 3.8.4. Differences between Hospital and Community Pharmacies

There are significant differences in working conditions between hospital (state-owned) and community (private) pharmacies. Hospital pharmacists generally enjoy higher salaries, less pressure, and more regular working hours, including fewer weekend shifts.


*“My mother completed her residency, so today she works at the oncology hospital pharmacy preparing cytostatics. She has a 6 h workday without weekends. The salary is better than in the private sector. It is a much more noble profession than working at the counter (NN in community pharmacy)”.*


#### 3.8.5. Impact of New Pharmaceutical Services

Introducing new pharmaceutical services such as flu vaccinations increased pharmacists’ workloads with minimal salary increases. This additional responsibility further strained pharmacists’ capacity to manage their primary duties.


*“Doctors are paid RON 40 (EUR 8) per flu shot administration. Pharmacies in the pilot program were not paid for administrations, so they probably took it out of the markup on the dose.”*


Some other studies indicate moderate satisfaction with pharmacy jobs associated with vaccination services, and the need for better reimbursement was also mentioned [17].

Analyzing workplace dynamics in the context of pharmacy professional satisfaction reveals a complex interplay of factors, including management practices, compensation structures, working conditions, and recognition. Addressing these issues requires a comprehensive approach involving transparent communication, fair compensation, and a supportive working environment that values the professional contributions of pharmacists. The insights from this research provide a foundation for developing strategies to enhance job satisfaction and retention within the pharmacy profession.

### 3.9. Limitations

While this study provides valuable insights into the factors influencing pharmacists’ professional satisfaction, several limitations should be acknowledged. First, netnography relies on data from online forums and social media, which may not fully represent the experiences of all pharmacists. Participants in these online communities may have different characteristics and concerns than those who do not engage in such platforms, potentially leading to selection bias. This limitation should be considered when interpreting the study’s findings, and future research could benefit from incorporating more diverse data sources. Additionally, the anonymity of online interactions might result in omitting critical contextual details, which could affect the depth and accuracy of the analysis.

Second, while offering rich, detailed insights, the study’s qualitative nature limits the generalizability of the findings. The themes identified are specific to the sample studied and may not apply universally to all pharmacists or settings. Moreover, the thematic analysis is inherently subjective, and despite rigorous coding and analysis procedures, researcher bias may influence the interpretation of the data. Future research could address these limitations by incorporating a mixed-methods approach, including quantitative surveys to validate and extend the findings, and expanding the sample to include a broader range of pharmacists from different regions and practice settings.

## 4. Conclusions

This study comprehensively examines the factors influencing professional satisfaction among pharmacists in Romania, utilizing a netnographic approach to gather and analyze data from online communities. The findings highlight several critical issues that impact pharmacists’ professional lives and well-being, including dissatisfaction with salaries, challenges related to professional recognition, and the demanding nature of their educational journey.

### 4.1. Key Findings

(a)Salary dissatisfaction: The data indicate widespread dissatisfaction with pharmacists’ salaries. Many believe that their compensation does not reflect the extensive training and critical responsibilities of their roles. This dissatisfaction is particularly noticeable among newly graduated pharmacists with low starting salaries and limited career prospects.(b)Professional recognition: Pharmacists often do not receive the recognition they deserve for their professional contributions. They are usually seen as salespeople rather than valued healthcare team members, which undermines their professional identity and leads to frustration and undervaluation.(c)Educational challenges: The study highlights the challenges often seen as unrelated content in the pharmacy curriculum. Many pharmacists say that their college education did not fully equip them for the practical requirements of their job, leading to a significant learning curve when they started working.(d)Organizational support and work environment: Effective organizational support is crucial for enhancing pharmacists’ job satisfaction. Poor management practices, inadequate facilities, and insufficient recognition can lead to stress and burnout. Creating supportive work environments and promoting work–life balance is essential for maintaining pharmacists’ well-being.(e)Technological and global trends: Technological advancements, especially AI, significantly impact the pharmacy profession, offering improved efficiency but posing challenges such as job redundancy. Global mobility and international practice comparisons highlight opportunities and disparities in the profession.

### 4.2. Recommendations

Based on the findings of this study, several key recommendations can be made to enhance pharmacists’ professional satisfaction and well-being.

Firstly, addressing salary dissatisfaction is paramount. Policymakers should implement measures to ensure fair and competitive compensation for pharmacists, reflecting their extensive training and critical role in healthcare. Establishing clear salary standards and supporting effective salary negotiation practices will help align compensation with pharmacists’ professional contributions.

Secondly, educational reforms are necessary to better prepare pharmacy students for the practical demands of their profession. Curricula should be updated to include relevant skills such as financial literacy, marketing, business management, and traditional medical knowledge. These changes will equip future pharmacists with the comprehensive skill set needed to thrive in a dynamic healthcare environment.

Thirdly, improving professional recognition is crucial for enhancing pharmacists’ job satisfaction. Efforts should be made to promote a better understanding of pharmacists’ roles as integral healthcare providers; this includes public awareness campaigns and initiatives within healthcare organizations to ensure pharmacists’ contributions are adequately recognized and valued.

Moreover, enhancing organizational support is essential for reducing burnout and fostering job satisfaction; this involves improving management practices, increasing staffing levels, and providing adequate facilities. Creating a supportive work environment and promoting work–life balance are critical steps towards maintaining pharmacists’ well-being and professional commitment.

Finally, integrating technological advancements, particularly AI, presents opportunities and challenges for the pharmacy profession. Continuous professional development and upskilling must ensure pharmacists can effectively adapt to and leverage these technologies. Preparing pharmacists for the future technological landscape will enable them to enhance their roles and improve healthcare delivery.

To sum up, by addressing these recommendations, stakeholders can significantly improve pharmacists’ professional lives and enhance their contributions to the healthcare system. These changes will benefit pharmacists and lead to better healthcare outcomes for patients. The insights from this study provide a valuable foundation for future research and policy development aimed at supporting the pharmacy profession and ensuring its sustainability in an evolving healthcare landscape.

## Figures and Tables

**Table 1 pharmacy-12-00155-t001:** Categories and codes derived from social media conversations on pharmacy professional satisfaction.

Category Name	Category Attribute	Category Description	Codes
Cultural Insights and Humor	Societal attitudes and informal expressions	Captures observations about societal attitudes toward pharmacy as a profession and informal expressions such as humor and sarcasm used by pharmacists.	Humor and sarcasm, social/cultural commentary
Economic and Financial Insights	Financial aspects influencing career decisions	Discusses the economic factors affecting pharmacists, including salary expectations, the impact of the cost of living, and broader financial outcomes related to the profession.	Economic and financial aspects, geographic differences
Global Trends and Technological Impacts	International influence and technological advancements	Captures the global trends affecting pharmacy practice and the impact of technological advancements such as AI on the profession	Implication of AI in pharmacy practice, international trends in pharmacy
Personal Experiences and Satisfaction	Individual experiences and job satisfaction	Reflects the personal experiences of pharmacists, encompassing career satisfaction, personal anecdotes, and emotional reactions to their professional roles.	Leaving the job, professional satisfaction and experiences
Professional Growth and Education	Development and educational pathways	Covers all aspects related to the educational background, continuous professional development, growth opportunities, and interdisciplinary collaboration within the pharmacy field	Alternative career paths, career development and opportunities, education and learning environment, interdisciplinary opportunities, international opportunities and language learning, lack of competition at admission
Professional Identity and Public Perception	Professional status and societal views	Examines how pharmacists are perceived within the healthcare community and by society, including misconceptions, stereotypes, and the level of professional recognition.	Compared to other professions, misconceptions and stereotypes, pharmacist as salesman, professional organization, professional recognition, social/cultural commentary
Regulatory and Market Environment	Legal, political, and market influences	Focuses on the regulatory, legal, and market-driven aspects impacting the pharmacy practice and business.	Pharmaceutical services, pharmacy market, politics, regulation on pharmacy setup, type of community pharmacy
Workplace Dynamics	Working conditions and the job market	Explores the dynamics of different working environments, challenges faced in the workplace, and the realities of the job market for pharmacists	Community vs. industrial pharmacy, job market reality, pharmacist as salesman, workplace conditions and challenges

**Table 2 pharmacy-12-00155-t002:** Summary of key findings in the “Cultural Insights and Humor” category from social media discussions among pharmacists.

Finding	Description
Challenges in choosing the profession	Humorously reflecting on the irony and difficulties of choosing a pharmacy career.
Correct spelling and precision	Emphasizing the importance of accuracy in the profession through humorous anecdotes.
Comparison with “Breaking Bad”	Comparing pharmacists’ skills to those depicted in the “Breaking Bad” series.
Leaving the job due to low salaries	Using humor to highlight dissatisfaction with salaries and subsequent career shifts to other fields.
Perceived inferiority compared to doctors	Expressing feelings of being undervalued compared to doctors and lacking public recognition, often articulated through humor.
Burnout and occupational stress	Addressing the issue of burnout and the overwhelming nature of the job with humorous remarks.
Sarcasm about financial transparency	Critiquing the financial strategies and perceived lack of transparency within pharmacy chains through sarcastic comments.
Overload with curricula during studies	Humor and sarcasm are used to question the topic’s relevance and to manage the overload of academic subjects in pharmacy education.
Redundancy in the age of automation	Highlighting the perceived redundancy of pharmacists’ tasks due to automation and technological advancements, often with a humorous or sarcastic tone.

**Table 3 pharmacy-12-00155-t003:** Summary of key findings in the “Economic and Financial Insights” category.

Finding	Description
Widespread dissatisfaction with salaries	Many pharmacists report dissatisfaction with their salaries, feeling that their compensation does not reflect their responsibilities and contributions to healthcare.
Low entry-level salaries for recent graduates	Newly graduated pharmacists face low starting salaries and limited career prospects, impacting their overall job satisfaction and retention.
Importance and challenges of salary negotiation	Effective salary negotiation is crucial for better compensation, but many pharmacists find it challenging to achieve favorable outcomes.
Job satisfaction and compensation	The relationship between job satisfaction and compensation is significant, with many pharmacists feeling under-compensated for their extensive training and critical responsibilities.
Comparison of salaries with other professions	Pharmacists often compare their salaries unfavorably with other professions that require similar or less educational investment, highlighting a perceived injustice in remuneration.
Career development paths	The opportunities for career advancement and the impact of additional certifications and specializations on earnings and professional growth are explored.
Market dynamics and indication of salary levels	The analysis highlights the influence of market dynamics on salary levels, including the impact of geographic location and market saturation on compensation.

**Table 4 pharmacy-12-00155-t004:** Summary of key findings in the “Global Trends And Technological Impacts” category from social media discussions among pharmacists.

Finding	Description
Appealing pharmacy models (Hungary and USA)	Different pharmacy models, such as those in Hungary and the USA, are perceived as more appealing due to higher salaries, minimum wage standards, and union negotiations.
Global mobility for pharmacists (UK and Ireland)	Pharmacists can move and work in other countries with just a language exam, with the UK and Ireland being notable examples.
Significant impact of AI on pharmacy work	AI is anticipated to significantly impact the work of pharmacists, potentially replacing many of their current tasks and raising concerns about job redundancy.

**Table 5 pharmacy-12-00155-t005:** Summary of key findings in the “Personal Experiences and Satisfaction” category from social media discussions among pharmacists.

Finding	Description
Working in profit-oriented chains and greedy managers	Pharmacists often encounter profit-oriented chains, and managers focus more on profits than on the well-being of their staff.
Low salaries and poor working conditions	Many pharmacists report low salaries and poor working conditions, contributing to job dissatisfaction.
Challenging work environment and stressful interactions	The work environment can be challenging, with stressful interactions with patients and management adding to job strain.
Better opportunities in alternative jobs	Some pharmacists find better job satisfaction and compensation in alternative careers outside traditional pharmacy roles.
Young pharmacists and career mobility	Young pharmacists demonstrate significant career mobility, often seeking better opportunities and conditions.
Salary disparities among pharmacists	There are notable salary disparities among pharmacists, influenced by location, experience, and type of employer.
Migration for better opportunities	Many pharmacists consider migrating to other countries for better professional and financial opportunities.
Challenges in academia and research	Pharmacists in academia and research face distinct challenges, including funding issues and career progression obstacles.
Passion and fulfilment in work	Despite the challenges, some pharmacists express strong passion and fulfilment.
Career options beyond community pharmacies	Pharmacists have various career options beyond community pharmacies, including industry roles, regulatory affairs, and clinical specialities.
Choosing pharmacy for interest in chemistry	Many pharmacists chose the profession due to a strong interest in chemistry and the sciences.
Difficulties for new graduates	New graduates often face significant difficulties, including finding suitable employment and transitioning from education to professional practice.

**Table 6 pharmacy-12-00155-t006:** Summary of key findings in the “Professional Growth and Education” category.

Finding	Description
Difficult university curriculum	Pharmacy students often face a rigorous curriculum with some subjects deemed unnecessary for their future careers.
Perceptions of pharmacy universities	There is a declining interest in pharmacy schools due to perceived low rewards and high demands.
Residency programs for better income	Residency programs provide an opportunity for additional income and allow pharmacists to balance multiple roles.
Career growth in industry and authorities	The pharmaceutical industry and regulatory bodies are areas for significant career advancement.
Going abroad for better prospects	Many pharmacists consider relocating to countries with better salary prospects and work conditions.

**Table 7 pharmacy-12-00155-t007:** Summary of key findings in the “Professional Identity and Public Perception” category.

Finding	Description
Changing the attitude of pharmacists related to the professional organization.	Establishing the own limits, refusing inappropriate working conditions and inadequate salary in the absence of support from the professional organization.
Disappointment about low salaries compared to other professions	Many other jobs, qualified but more unqualified, offer a salary comparable to that of a pharmacist or much higher.
Misconceptions: pharmacists are salespeople, not healthcare providers	People have many wrong perceptions and stereotypes regarding the pharmacist profession, respectively, the pharmacy as an institution. Due to the commercial character of the pharmacy, the pharmacist receives the seller’s label, although the main attributions are to advise patients and to prepare medicines.
Frustrations about the lack of recognition of the profession’s status.	In the pharmacists’ guild, many frustrations accumulated regarding the profession’s appreciation and the inequity of other occupations in the medical field (e.g., doctors and dentists).
Pharmacists’ concerns: corporate takeover affects the professional image	The expansion of pharmaceutical chains negatively impacted the image of pharmacists; pharmacies are no longer entirely owned by pharmacists, leading to a decline in the number of independent pharmacies; independent pharmacies struggle to compete with large pharmaceutical chains.
The pharmacy profession is widely respected	A part of community pharmacies, other pharmacy specialities offer satisfaction to employees.

**Table 8 pharmacy-12-00155-t008:** Summary of Main Findings in the Regulatory and Market Environment.

Finding	Description
Ownership and market consolidation	Discusses the preference for sole ownership of pharmacies, market dominance by chains, vertical integration, and market saturation.
Regulations on pharmacist presence	Examines the need for differentiated regulations for larger pharmacies and additional compensation for chief pharmacists.
Taxes and economic regulations	Addresses the high tax burden on salaries and the economic challenges of the current business model for community and hospital pharmacies.
Labor regulations and professional pressures	Highlights the need for more explicit labor regulations, especially concerning work hours and the limited influence of professional organizations.
Market realities and pharmacist availability	Explores the disparities in pharmacist availability and salary differences between small towns and large cities.
Practice standards and commercial focus	Discusses the need to shift focus from commercial interests to professional and patient-centered care, including potential government support.

**Table 9 pharmacy-12-00155-t009:** Summary of main findings in the Workplace Dynamics category.

Finding	Description
Difficult working conditions	Hectic schedules, long hours, weekend work, pressure from management and patients, low salaries, and legal issues.
Better working conditions in smaller towns	Higher salaries, fewer weekend shifts, more flexibility, and reluctance to relocate.
Opportunities in the pharmaceutical industry	Better conditions and salaries in regulatory affairs, quality, clinical trials, sales, and marketing.
Differences between hospital and community pharmacies	Higher salaries, less pressure, fewer weekend shifts in hospitals, and challenging conditions in community pharmacies.
Impact of new pharmaceutical services	Increased workload with minimal salary increases due to new services such as flu vaccinations.

## Data Availability

The authors will make the raw data supporting this article’s conclusions available upon request.

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
