# Peer review of "Pharmacists’ Professional Satisfaction and Challenges: A Netnographic Analysis of Reddit and Facebook Discussions"

_pharmacy, 2024, doi:10.3390/pharmacy12050155_

Round 1

Reviewer 1 Report

Comments and Suggestions for Authors

Thank you for the opportunity to review this interesting, innovative, well conducted and well presented study.  Although dealing with only one country, the findings are appropriately compared to studies in other countries.  

In keeping with the methodology used, it is a relatively small sample of data, but rigorously analysed.  The themes are well validated and the exemplars are appropriate. 

It would assist international audiences however if the quoted salaries and rewards were also expressed in another more internationally familiar currency such as Euros or US$.

The findings are supported by the data, well discussed and the conclusions and recommendations are appropriate.

I believe this study will provoke quite a lot of discussion aboutthe direction of the profession in a number of countries.

Comments on the Quality of English Language

The English is mostly excellent with very minor editing required. 

Author Response

Dear Reviewer

Thank you very much for reviewing this paper and providing feedback.

Comment: It would assist international audiences however if the quoted salaries and rewards were also expressed in another more internationally familiar currency such as Euros or US$.

Response: The new version include figures also in Euro. The local currency (lei, RON) is quoted at 5 RON to 1 Euro.

Reviewer 2 Report

Comments and Suggestions for Authors

Easy to read manuscript although a bit long.  The findings and conclusions seem to be intuitive. Therefore, I am wondering you have anything uexpected to report?

I've attached a pdf of the manuscript that contains a few comments.

Author Response

Dear Reviewer,

Thank you very much for your effort to read and review our work. We appreciate a lot your comments.
The precise answers:

Comment 1: (1) what is $US equivalent of 2050 RON and (2) was is the relationship between RON and lei [mentioned in line 361 and following]

Response 1: One Euro is equivalent to five RON/lei, while one US dollar is equivalent to 4.5 RON/lei. The Romanian currency is called Leu (plural Lei) and its symbol is RON (Romanian New Leu). In social media comments, both forms are used, and we have retained the original text for comments. However, in order to make it easier for readers, we have included the equivalence of any amount in Euros in the new version of the article.

Comment 2: Did you find anything that was unexpected?

Response 2: We encountered several unexpected findings in our study:

  • public misconceptions: the extent of public perception of pharmacists as mere "salespeople" was deeper than anticipated.
  • concerns over automation: anxiety regarding the impact of automation on pharmacists' roles was higher than expected.

Comment 3: Why are some of the journal title in ALL CAPS?

Response 3: We used Mendeley Reference Manager for our references and imported them with it. In some cases, the titles were written in ALL CAPS. However, we have corrected this issue in the new version of the article. Thank you for bringing this to our attention.